# The Solid-State Synthesis of BiOIO_3_ Nanoplates with Boosted Photocatalytic Degradation Ability for Organic Contaminants

**DOI:** 10.3390/molecules28093681

**Published:** 2023-04-24

**Authors:** Jia Li, Jing Xie, Xiaojing Zhang, Enhui Lu, Yali Cao

**Affiliations:** State Key Laboratory of Chemistry and Utilization of Carbon Based Energy Resources, College of Chemistry, Xinjiang University, Urumqi 830046, China

**Keywords:** solid-state chemical reaction, BiOIO_3_, photocatalysis, degradation contaminant

## Abstract

BiOIO_3_ exhibits excellent oxidation capacity in the photocatalytic degradation of contaminants thanks to its unique polarized electric and internal electrostatic field. However, the synthetic method of BiOIO_3_ nanomaterials is mainly focused on hydrothermal technology, owing to its high energy consumption and time-consuming nature. In this work, a BiOIO_3_ nanosheet was prepared by a simple solid-state chemical reaction, which was identified by XRD, EDS, XPS, and HRTEM. Benefiting from the strong oxidation ability of the valence band maximum, the distinctive layer structure, and the promoted generation of ·O_2_^−^, the BiOIO_3_ nanosheet exhibits excellent photo-degradation activity for methyl orange (MO) and its apparent rate constant is 0.2179 min^−1^, which is about 3.02, 8.60, and 10.26 times higher than that of P25, BiOCl, and Bi_2_O_2_CO_3_, respectively. Interestingly, the BiOIO_3_ nanosheet also has good photocatalytic degradation performance for phenolic compounds; in particular, the degradation rate of BPA can reach 96.5% after 16 min, mainly due to hydroxylation reaction.

## 1. Introduction

With the rapid development of industry and accelerated socialization, the problem of environmental pollution, especially water pollution, has posed a serious threat to the survival of human beings. Thus, with the increasingly serious water pollution, there is an urgent need to find an environmentally friendly technology to remove toxic and harmful substances. Compared with the physical adsorption, biodegradation, and electrochemical methods, photocatalytic degradation technology has attracted much attention as it can completely decompose dyes into non-toxic chemicals (such as H_2_O and CO_2_), which has the advantages of easy operation, being environmentally friendly, and low energy consumption; more importantly, it can use solar energy directly [1,2,3]. In the reaction of photocatalytic degradation of organic pollutants, the photocatalyst is the core of the whole technology. As a traditional photocatalytic material, the low separation efficiency of the photogenerated electron–hole of TiO_2_ limits its further application [4,5]. Therefore, the development of new photocatalytic materials or the modification of traditional materials has become the focus of current research in the field of photocatalysis.

Layered bismuth-based photocatalysts characterized by (Bi_2_O_2_)^2+^ layers and interlayer ions or polyhedra as crystal structures, such as BiOX (X = Cl, Br, I), Bi_2_O_2_CO_3_, and Bi_2_XO_6_ (X = W, Mo), have been widely researched because their conduction bands have a positive potential, which has powerful oxidation ability for pollutants, and their special layered structure is conducive to the separation of carriers [6,7,8]. BiOIO_3_ is a novel Aurivillius-type layered bismuth-based compound; its crystal structure consists of (Bi_2_O_2_)^2+^ and (IO_3_)^−^ layers stacked alternately along the c-axis direction [9,10]. The unique polarized electric field of (IO_3_)^−^ and internal electrostatic field lead to the rapid separation of photogenerated electrons and holes. Thus, BiOIO_3_ exhibits an extremely strong catalytic oxidation ability and excellent photocatalytic activity in the degradation of pollutants. In 2013, Wang et al. [11] investigated the photocatalytic performance of BiOIO_3_ for the first time, finding that the catalytic degradation of methyl orange (MO) over BiOIO_3_ under UV light irradiation was significantly more active than TiO_2_. Regrettably, similar to TiO_2_, the band gap of BiOIO_3_ is also about 3.0 eV, which is the typical characteristic of UV-responsive photocatalysts [12,13]. Nowadays, to achieve a high conversion efficiency of sunlight to chemical energy, various strategies such as element doping or heterojunction construction have been developed to narrow the band gap of BiOIO_3_ photocatalysts. Liu et al. [14] synthesized nickel-doped BiOIO_3_ nanosheets using a hydrothermal method, which can increase the oxygen vacancy concentration, promote the separation of photo-carriers, improve the activation of molecular oxygen, and decrease the band gap. Hence, the photocatalytic removal rate of flue gas mercury by Ni-ion-doped BiOIO_3_ was 81.2%, which was about 1.4 times higher than that of pure BiOIO_3_. Huang et al. [15] constructed the I-doped BiOIO_3_ nanosheets using a hydrothermal method. Compared with pure BiOIO_3_, the photoresponse range of I-BiOIO_3_ was greatly extended from UV to visible light, thus photocatalysis experiments showed that the photocatalytic activity for the degradation of methyl orange and rhodamine B dyes over the I-doped BiOIO_3_ was significantly enhanced under visible light irradiation. Lai et al. [10] synthesized BiOIO_3_ loaded by carbon quantum dots (CQDs/BiOIO_3_) through a hydrothermal method. Here, 3 wt% CQDs/BiOIO_3_ with lamellar morphology exhibited a maximum photodegradation efficiency of 95.01% of the bisphenol A in 60 min. However, the main synthesis technique of BiOIO_3_ nanomaterials is the hydrothermal method at present; the hydrothermal temperature and the time should be at least 140~150 °C and 5~10 h, respectively [16,17]. As a consequence, the development of an easy-to-operate method with mild reaction conditions for the preparation of nanometer BiOIO_3_ photocatalysts is urgently needed.

Herein, a simple and easy-to-operate solvent-free chemical reaction, and subsequently a heat treatment process were used to prepare the BiOIO_3_ nanosheets. The effects of different temperatures and atmospheres on the phase composition, morphology, and energy band structure of the sample were investigated. The photocatalytic degradation performance of BiOIO_3_ on simulated dyes and phenolic pollutants was studied. The mechanism of photocatalytic degradation of MO over BiOIO_3_ nanosheets was clarified using a free radical capture experiment. The degradation process of BiOIO_3_ on bisphenol A was investigated using liquid chromatography mass spectrometry.

## 2. Results and Discussion

Figure 1a shows the XRD spectra of the reaction raw materials, precursor, and BiOIO_3_. Although the XRD spectra of the precursor cannot be indexed in the database, its diffraction peaks are completely different from those of the Bi(NO_3_)_3_·5H_2_O and KIO_3_, indicating that the powder of Bi(NO_3_)_3_·5H_2_O and KIO_3_ did undergo chemical reactions in the low-heating solid-state reaction. The diffraction peaks of the substance are consistent with the standard card of the orthorhombic crystal BiOIO_3_ (ICSD 262019) [11] and no other peaks of impurity were detected. Moreover, the high intensity of the (121) crystal plane for the diffraction spectrum of BiOIO_3_ indicates the good crystallinity of BiOIO_3_, and the preferential growth direction of BiOIO_3_ is the (121) direction. In addition, in order to further determine the composition of the precursor and the calcination temperature, the phase composition of the samples acquired from the calcining precursor at 200–500 °C in a muffle furnace was analyzed. As shown in Figure 1b, when the calcination temperature was 200 °C, the sample was a mixture of BiOIO_3_ and Bi_x_I_y_, and when the calcination temperature increased to 300 °C or 400 °C, both of the samples were pure BiOIO_3_. Continuing to increase the calcination temperature to 500 °C, the diffraction peaks of the sample were in full agreement with the standard card of orthorhombic Bi_5_O_7_I (JCPDS 40-0548), and the composition changed from BiOIO_3_ to Bi_5_O_7_I. This change can be attributed to the fact that iodide ion is unstable at a high temperature and some of the iodine would escape from the BiOIO_3_ crystals, causing the rearrangement of the Bi, O, and I atoms, and finally changing the physical phase. According to the variation in phase composition with the continuously increasing temperature, we can infer that the precursor is an iodine-rich bismuth mixture, from which the excess iodine sublimates at a high temperature and the remained Bi, I, and O atoms undergo reorganization, thus producing pure BiOIO_3_ at 300 °C and 400 °C. To further investigate the effect of the calcination atmosphere on the physical phase of the sample, the precursor was calcined in a neutral atmosphere (N_2_) and reducing atmosphere (the mixture of N_2_ and H_2_) at 300 °C with the same heating rate and holding time. As displayed in Appendix A, compared with pure BiOIO_3_ obtained in air, the diffraction peaks of the sample calcined in the N_2_–H_2_ mixture are completely consistent with BiOIO_3_, while the diffraction peaks of the sample achieved in the N_2_ atmosphere have impurity peaks appearing at 26° and 43°.

To provide direct evidence to confirm the coexistence of the Bi, O, and I elements in BiOIO_3_, an energy-dispersive spectrometer (EDS) was further employed to demonstrate BiOIO_3_ derived at 300 °C in an air atmosphere. As presented Figure 1c, the EDS spectrum pattern revealed that the Bi, I, and O elements had atomic ratio of 13.21:16.50:70.29, which is basically consistent with the composition of the BiOIO_3_ product. In addition, XPS was also used to characterize the surface chemical composition and chemical states of BiOIO_3_, with a full elemental scan showing that only Bi, I, O, and a trace of C species were detected (Figure 1c), while the C peak is from adventitious hydrocarbon on the XPS instrument. The high magnification XPS of Bi 4f has two peaks at binding energies of 159.08 eV and 164.38 eV, corresponding to Bi 4f_7/2_ and Bi 4f_5/2_, respectively (Figure 1e), which can be ascribed to the trivalent bismuth ion. The peak of I 3d can be divided into two peaks at binding energies of 623.78 eV and 635.18 eV, matching well with I 3d_5/2_ and I 3d_3/2_ (Figure 1f), respectively, of I^5+^. Both of them further evidence that pure BiOIO_3_ can be successfully fabricated via a two-step solid-phase chemical reaction.

The morphologies of the precursor and BiOIO_3_ were characterized by FESEM, TEM, and HRTEM. As exhibited in Appendix A, the precursor is of homogeneous granular shapes with diameters of about 20–50 nm. SEM and TEM show that BiOIO_3_ has a two-dimensional lamellar structure (Figure 2a–c), with these irregular nanosheets stacked together with a size of about tens of nanometers to submicron and thicknesses of about 28 nm. The nanoplate structure may because the (Bi_2_O_2_)^2+^ and (IO_3_)^−^ layers in the crystal structure of BiOIO_3_ are alternately arranged, thus easily forming a nanosheet morphology (Appendix A). As observed from HRTEM, the spacing between the adjacent lattice stripes is 0.324 nm (Figure 2b), corresponding to the (121) plane of the orthorhombic BiOIO_3_ crystal, further illustrating the successful synthesis of pure BiOIO_3_.

In Figure 3a, the N_2_ gas adsorption–desorption isotherm of BiOIO_3_ is measured and type IV with an H3 hysteresis loop was detected, implying the mesopore structure of the photocatalysts. The pore size distribution demonstrated that many mesoporous structures of about 15–25 and 30–45 nm coexist in BiOIO_3_. The Brunauer-Emmett-Teller (BET) surface area of BiOIO_3_ calculated from the N_2_ isotherms is approximately 8.932 m^2^/g, suggesting the specific surface area is not the main factor affecting performance. 

The optical properties of the prepared sample were measured by UV/Vis diffuse reflectance spectroscopy. As can be seen from Figure 3b, BiOIO_3_ has strong absorption intensities in the region of UV and the absorption edge is about 390 nm. The band gap (Eg) can be fitted by the equation αhν = A(hν − Eg)^n/2^ [18], where α is the absorption coefficient, hν is photon energy, A is the constant, and the value of n depends on the type of semiconductor. As presented in the insert of Figure 3b, the band gap (Eg) of BiOIO_3_ is 3.3 eV, further manifesting BiOIO_3_ as an ultraviolet-light-responsive photocatalyst.

The photocatalytic activity of BiOIO_3_ nanosheets under UV light was evaluated using anionic dyes (methyl orange) and cationic dyes (rhodamine B and methylene blue) as simulated pollutants, and the concentration of all contaminants is 10 mg/L. Firstly, the effect of calcination atmosphere on the photocatalytic removal performance was investigated; the results are illustrated in Appendix A. Although the performance of the sample obtained under the reducing atmosphere was better than that of the other two samples during dark adsorption, the degradation rate of BiOIO_3_ calcined in the oxidizing atmosphere (in the air) was the fastest. Therefore, the sample BiOIO_3_ obtained under an air atmosphere was used for further study. 

In addition, the effect of reaction conditions (catalyst concentration and pH) on the photocatalytic performance was also explored. Appendix A shows the degradation curve of MO by different concentrations of BiOIO_3_ under UV irradiation. It can be observed that the adsorption capacity is unchanged, while the degradation rate increases with the increase in the catalyst concentration. The degradation efficiency of dye molecules is about 48% and 86% after irradiation for 15 min when the concentration of the photocatalyst in the system is 0.2 g/L and 0.4 g/L, respectively. When the concentration of the photocatalyst in the system is 0.6 g/L or 0.8 g/L, MO is completely degraded when exposed to light for 12 min. Appendix A presents the degradation efficiencies of MO over BiOIO_3_ at different pH. The apparent rate constants in a neutral environment (pH = 7) are higher than in acidic and alkaline environments, illustrating that neutral (pH = 7) environments are more suitable for the degradation of MO over BiOIO_3_. Therefore, the following catalytic experiment was conducted in a neutral environment (pH = 7) with a catalyst concentration of 0.2 g/L. As presented in Figure 4a, the dark adsorption of BiOIO_3_ for MO, RhB, and MB is almost the same and negligible. However, the degradation efficiency of BiOIO_3_ nanosheets for MO, RhB, and MB varies widely after 16 min of UV light irradiation, and the removal rates are 100%, 85.2%, and 65.2%, respectively. Figure 4b demonstrates that the characteristic absorption peak of methyl orange gradually decreases with the increase in light irradiation time, and its color also gradually changes from yellow to colorless. As seen in Figure 4c, the photocatalytic activity of BiOIO_3_ nanosheets was compared with other photocatalysts of bismuth oxygen compound (BiOCl and Bi_2_O_2_CO_3_) synthesized by room-temperature solid-state chemical reaction, commercial TiO_2_ (P25), and blank (no photocatalyst), using MO as the dye. After the adsorption and desorption equilibrium, there was basically no adsorption of methyl orange over the other samples, except for Bi_2_O_2_CO_3_. Although the adsorption efficiency of Bi_2_O_2_CO_3_ on MO was as high as 70%, MO is quite recalcitrant to degrade by Bi_2_O_2_CO_3_ under UV light illumination. Therefore, the photocatalytic degradation kinetics of MO was investigated using different photocatalytics, and the apparent rate constants for the degradation of MO by BiOIO_3_ nanosheets were fitted to 0.2179 min^−1^ under UV light (Figure 4d), which were about 3.02, 8.60, and 10.26 times higher than those of P25 (0.0721 min^−1^), BiOCl (0.0253 min^−1^), and Bi_2_O_2_CO_3_ (0.0212 min^−1^), respectively. Besides, the degradation efficiency for MO over BiOIO_3_ is in agreement with other reported photocatalysts. The above results indicate that BiOIO_3_ has superior photocatalytic activity compared with other bismuth-based materials and P25.

In order to evaluate the repeatability and stability of the photocatalyst, a recycling experiment was conducted on the UV photocatalytic degradation of MO. As depicted in Figure 4e, there is almost no significant decrease in BiOIO_3_ nanosheets during five cycles of the degradation of MO. Moreover, the XRD pattern and the relative peak intensity of the BiOIO_3_ photocatalyst after five cycles remain unchanged as compared with the fresh catalyst (Figure 4f), and no BiOI or other impurities appeared, indicating that BiOIO_3_ nanosheets are stable in the photocatalytic reaction. 

To explore the mechanism of photooxidation reactions for the decomposition of MO, capture experiments on the main active free radical were carried out. Benzoquinone (BQ), tertiary butanol (TBA), and ethylenediamine tetraacetic acid disodium salt (EDTA-2Na) were applied to capture ·O_2_^−^, ·OH, and h^+^, respectively [7,19]. As displayed in Figure 5a, the photocatalytic degradation efficiency of BiOIO_3_ was greatly inhibited by BQ, illustrating that ·O_2_^−^ is the main prominent active free radical for removing MO. The addition of EDTA-2Na has some effect on the decomposition, indicating that h^+^ plays an auxiliary role in photocatalysis, while the inexistence of ·OH has no influence on the photocatalytic activity of BiOIO_3_ in eliminating MO. In addition, the trapping agent DMPO was used to detect ·OH in water and ·O_2_^−^ in methanol on ESR to further confirm the active radical ·OH and ·O_2_^−^. As shown in Figure 5b, there are no any signals in dark, while obviously characteristic signals of DMPO-·OH and DMPO-·O_2_^−^ were observed after UV light irradiation and the intensity of the DMPO-·O_2_^−^ signal increased with the prolonged irradiation time, offering conclusive evidence for the increased generation of ·O_2_^−^. The above result indicates that ·O_2_^−^ is the major active radical and the oxidation reaction played a leading role for the degradation of MO. 

Moreover, the potentials of the valence band (VB) and conduction band (CB) of BiOIO_3_ can be calculated with the two formulas proposed by Butler and Ginley [20,21]: E_VB_-E_e_ + 0.5 E_g_ and E_CB_ = E_VB_ − E_g_, where E_VB_ and E_CB_ are the VB potential and CB potential of the as-prepared material, respectively; E_e_ is the energy of free electrons on the hydrogen scale (ca. 4.5 eV vs. NHE), and E_g_ is the absolute electronegativity of the corresponding semiconductor material. Based on the above formulas and the optical absorption spectrum, E_VB_ and E_CB_ are +3.99 and +0.69 eV, respectively, indicating the strong oxidation ability of the valence band maximum. As displayed in Figure 5c, the reduction potentials of O_2_/·O_2_^−^ (−0.046 eV vs. NHE) are more negative than the conduction band minimum of BiOIO_3_; therefore, ·O_2_^−^ can be generated only when the light is below 308 nm, further manifesting that the degradation of MO over BiOIO_3_ is largely dependent on the oxidation of ·O_2_^−^.

To prove the activity of the BiOIO_3_ photocatalyst for various organic contaminants, photocatalytic degradation of phenol and phenolic compounds was conducted. Phenol and phenolic compounds are a class of organic pollutants that are highly toxic, carcinogenic, and persistent organic pollutants, which are widely used in plastics and epoxy resin manufacturing. As presented in Figure 6a–d, phenol (ph-OH), 2,4-dichlorophenol (2,4-DCP), tetracycline hydrochloride (TC), and bisphenol A (BPA) were partially or completely degraded by BiOIO_3_ nanosheets when all of their concentrations were 10 mg/L. In particular, the characteristic absorption peaks of TC and BPA decreased sharply, suggesting the extensive removal ability of BiOIO_3_. The degradation efficiency of ph-OH, 2,4-DCP, TC, and BPA by BiOIO_3_ nanosheets was 22.6%, 29.5%, 85.7%, and 96.5%, respectively, after 16 min of light exposure (Figure 6e). To further investigate the intermediates of the photocatalytic process, the residual liquid for the degradation of bisphenol A after 12 min was tested by liquid chromatography coupled with mass spectrometry (LC-MS); the results are shown in Table 1. Five mass spectrometric peaks with different retention times were present in the detection results; the mass-to-charge ratios (*m*/*z*) of these intermediates were *m*/*z* = 163.0389 ([M-H]^−^), *m*/*z* = 259.0965 ([M-H]^−^), *m*/*z* = 133.0647 ([M-H]^−^), *m*/*z* = 151.0753 ([M-H]^−^), and *m*/*z* = 243.1022 ([M-H]^−^), which were analyzed as C_9_H_8_O_3_, C_15_H_16_O_4_, C_9_H_10_O, C_9_H_12_O_2_, and C_14_H_14_O_4_, respectively [22,23]. Based on the above analysis and the reported literature, the degradation of BPA is mainly involved the reaction of hydroxylation; the possible degradation pathway is speculatively shown in Figure 6f [24,25].

## 3. Materials and Methods

### 3.1. Preparation of BiOIO_3_

In a typical solid-state chemical reaction, 5 mmol (2.425 g) of bismuth(III) nitrate pentahydrate (Bi(NO_3_)_3_·5H_2_O) and 10 mmol (2.14 g) of potassium iodate (KIO₃) were separately weighed and ground in an agate mortar for about 10 min. The two reactants remained as powders after contact and grinding, and no obvious reaction was observed. After continuous grinding for 30 min, the acquired mixture was packaged in a 50 mL conical flask and placed into a 60 °C water bath for 2 h to ensure the chemical reaction was complete. Subsequently, a white precursor was obtained by washing and filtering with deionized water, and then dried in air at room temperature. The precursor was held in a muffle furnace at 300 °C for 2 h with a heating rate of 3 °C/min to obtain a final white sample, which was denoted as “BiOIO_3_”. The synthetic procedure of the other samples was the same as above.

The reaction equation is deduced [26] and shown as follows:Bi(NO_3_)_3_·5H_2_O + KIO_3_→BiOIO_3_ + KNO_3_ + 4H_2_O + 2HNO_3_(1)

### 3.2. Characterization

X-ray diffraction (XRD) was used to analyze the crystalline phase and chemical constitution of the sample by a Bruker D8 using filtered Cu Kα radiation (1.54056 Å) with an operating voltage of 40 kV and a beam current of 40 mA. X-ray photoelectron spectra (XPS) were recorded using the ESCALAB 250 Xi system with monochromatic Al Kα radiation (1486.6 eV); the binding energy was calibrated by a C 1s line at 248.11 eV, 284.25 eV, and 284.75 eV. The oxygen defects were recorded by an in situ electron paramagnetic resonance (EPR, JES FA300) spectrometer at 77 K in liquid nitrogen. Field emission scanning electronic microscopy (FESEM, SU8010, 5 kV) and transmission electron microscopy (TEM, JEOL JEM-2100F) were performed to obtain the morphology and size of the samples. The elementary composition was performed on an energy-dispersive X-ray spectrometer (EDS, Oxfold 2000) connected to an SEM. The specific surface area of the sample was obtained using the Brunauer-Emmett-Teller (BET) method (Quantachrome, AutosorbiQ2). The UV/Vis diffuse reflectance spectroscopy was conducted on a Hitachi U-3010 spectrophotometer using BaSO_4_ as the reference. The UV/Vis absorption spectra of as-generated adsorbents were obtained using a Hitachi U-3900H spectrophotometer during 200~800 nm. The intermediate products were monitored by liquid chromatography mass spectrometry (LC-MS, Thermo Fisher Scientific Ultimate 3000/Q-Exactive, Waltham, MA, USA). The electron spin resonance (ESR) signals were measured on a JES FA300 spectrometer in 5,5’-dimethyl-1-pirroline-N-oxide (DMPO) solutions (methanol dispersion for DMPO-O_2_**^−^** and distilled water dispersion for DMPO-OH).

### 3.3. The Degradation Performance Test

Photocatalytic performance test: the XPA-1 photochemical reactor (Xujiang Electromechanical Plant, Nanjing, China) with a 300 W mercury lamp as the UV light source was employed for the photocatalytic degradation experiments. The light source radiates the reaction liquid through a glass cold trap with condensed water. The main contaminants selected for simulation are methyl orange (MO), methylene blue (MB), and rhodamine B (RhB). The procedure was as follows: 25 mg of photocatalyst was ultrasonically dispersed in 50 mL of organic dye and subjected to a dark reaction for 30 min to achieve physical adsorption–desorption equilibrium. Then, light irradiation was performed and 4 mL of sample was taken at 5 min intervals during the reaction; subsequently, the supernate was separated by centrifugation and the absorbance of the clear solution was measured using a UV/Vis spectrophotometer.

Cyclic test: After testing the photocatalytic performance of the samples using the above method, the solid photocatalytic material was separated from the mixture by centrifuging, washing, and drying. Afterwards, the recovered photocatalyst was used again to test the photocatalytic performance with three repetitions.

## 4. Conclusions

In this paper, BiOIO_3_ nanosheets were firstly prepared using an easy-to-handle and low-cost two-step solid-state chemical reaction. Thanks to the unique layered crystal structure of BiOIO_3_, it demonstrates superior photocatalytic degradation activity as well as better repeatability and cycling stability for cationic dye MO than other layered bismuth-based materials (BiOCl and Bi_2_O_2_CO_3_) and P25 under ultraviolet light, which can completely degrade MO within 10 min. Besides, BiOIO_3_ nanosheets can also be employed to degrade anionic dyes RhB and MB and neutral pollutants ph-OH, 2,4-DCP, BPA, and TC. Unexpectedly, the removal of MO over BiOIO_3_ is largely dependent on the oxidation of ·O_2_^−^, while the hydroxylation reaction is the main reason for the degradation of BPA. The study may not only introduce a new simple approach to synthesize BiOIO_3_ nanomaterials, but also provide a candidate with high photocatalytic activity for environmental purification.

## Figures and Tables

**Figure 1 molecules-28-03681-f001:**
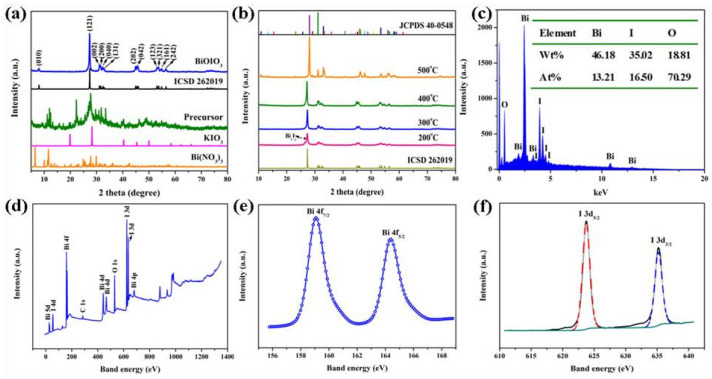
XRD patterns of (**a**) the precursor and BiOIO_3_ and (**b**) the samples obtained by the calcining precursor at different temperatures; (**c**) the EDS result of the BiOIO_3_. XPS spectra of BiOIO_3_: (**d**) survey scan, (**e**) Bi 4f, and (**f**) I 3d.

**Figure 2 molecules-28-03681-f002:**
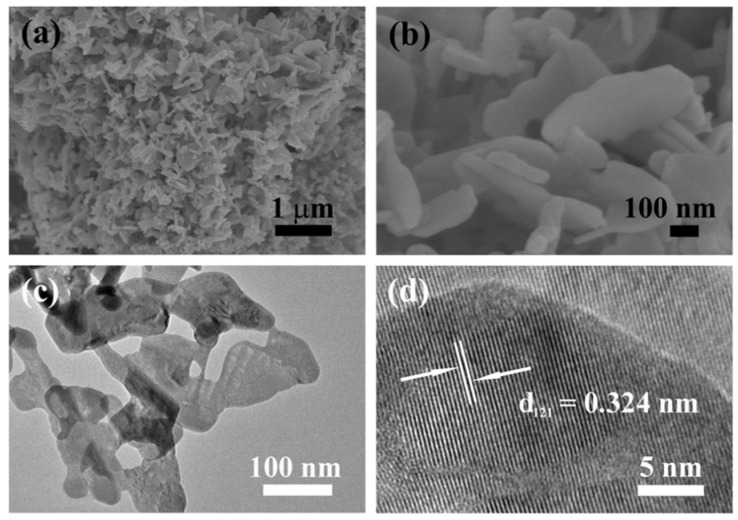
(**a**,**b**) FESEM images, (**c**) TEM, and (**d**) HRTEM of BiOIO_3_.

**Figure 3 molecules-28-03681-f003:**
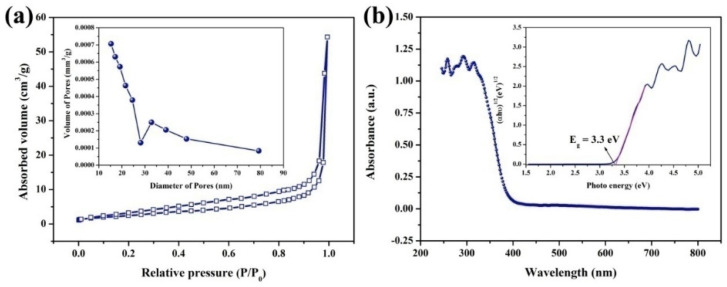
(**a**) The nitrogen gas absorption–desorption curve and pore size distribution and (**b**) UV/Vis diffuse reflectance spectra and band gap of BiOIO_3_.

**Figure 4 molecules-28-03681-f004:**
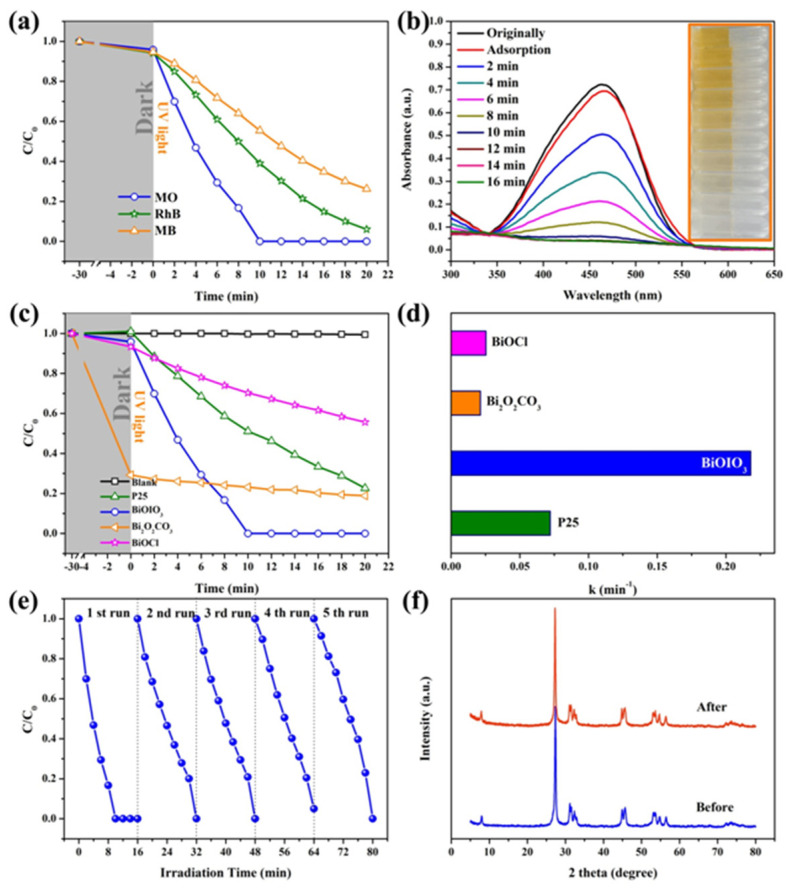
(**a**) Degradation efficiencies of MO, RhB, and MB over BiOIO_3_ nanoplates under UV light irradiation; (**b**) UV/Vis spectral change of MO with irradiation time over BiOIO_3_ nanoplates; (**c**) degradation efficiencies of MO over BiOIO_3_, BiOCl, Bi_2_O_2_CO_3_, and P25 under UV light irradiation; (**d**) apparent first-order rate constants for BiOIO_3_, BiOCl, Bi_2_O_2_CO_3_, and P25 photocatalysts; (**e**) recycling experiments conducted on the UV photocatalytic degradation of MO by BiOIO_3_ nanoplates; and (**f**) XRD pattern of before and after photocatalysis for BiOIO_3_ nanoplates.

**Figure 5 molecules-28-03681-f005:**
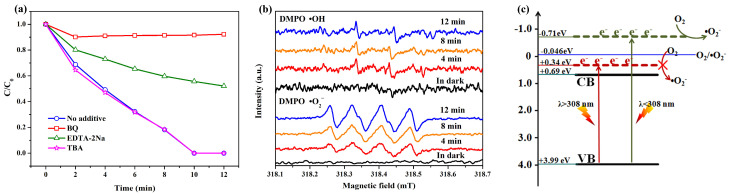
(**a**) Photocatalytic degradation of MO over BiOIO_3_ nanoplates together with different radical scavengers. (**b**) ESR spectra of BiOIO_3_ nanoplates for the detection of ·OH and ·O_2_^−^. (**c**) The photocatalytic degradation mechanism of BiOIO_3_ nanoplates.

**Figure 6 molecules-28-03681-f006:**
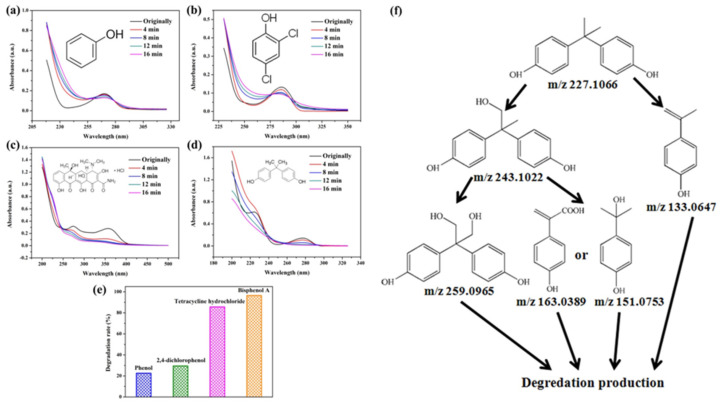
Temporal UV/Vis spectral change of (**a**) ph-OH, (**b**) 2,4-DCP, (**c**) TC, and (**d**) BPA with irradiation time over BiOIO_3_ nanoplates; (**e**) the degradation rate of different pollutants after 16 min of UV light over BiOIO_3_ nanoplates; and (**f**) the possible degradation pathway for BPA.

**Table 1 molecules-28-03681-t001:** The ions *m*/*z* and molecular structures of intermediate products at different retention times.

Peak NO.	Retention Time (min)	Detected Ions *m*/*z*	Molecular Structure
1	1.28	163.0389	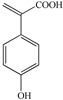
2	1.92	259.0965	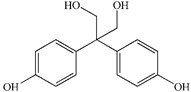
3	1.99	133.0647	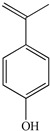
4	1.99	151.0753	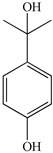
5	2.90	243.1022	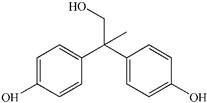
6	4.00	227.1066	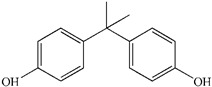

## Data Availability

Not applicable.

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
