# Peer review of "The Solid-State Synthesis of BiOIO3 Nanoplates with Boosted Photocatalytic Degradation Ability for Organic Contaminants"

_molecules, 2023, doi:10.3390/molecules28093681_

Round 1
Reviewer 1 Report
Dear Editor
This study investigated a simple approach to the synthesis of BiOIO3 nanomaterials using boosting photocatalytic with high photocatalytic activity for environmental purification.
1. Eq.1 needs a reference
2. Conclusion should be improved
3. The Brunauer-Emmett-Teller (BET) surface area of BiOIO3 calculated from N2 isotherms is approximately 8.932 m2/g, suggesting the specific surface area is not the main factor affecting performance…compare the pore size of BiOIO3 that synthesized with other approaches.
4. Quality of Fig.5 needs to be improved
Reviewer 2 Report
In this manuscript, Li and co-workers studied BiOIO3 nanosheets application in the photocatalytic degradation of methyl orange (MO). The authors found that the prepared material could degrade about 100% of the dye after 16 min. On the other hand, there are several issues which should be solved before publication in Molecules:
1. There are a few formatting and spelling mistakes, which should to be corrected.
2. In Introduction, the authors indicated that wide band gap of TiO2 limits its practical application. However, the band gap of the BiOIO3 nanosheets is about 3.3 eV according to UV-Vis diffuse reflectance spectra. So, it is necessary to explain benefits of the proposed material more carefully.
3. The novelty of the manuscript should be diclosed in more details due to a huge number of the papers devoted to the use of BiOIO3 in photocatalysis have been published.
4. The authors should investigate the effect of reaction conditions (catalyst and pollutant concentration, pH, etc.) on the photocatalytic performance.
5. A table, where a comparison of this study with other investigations, should be included. This could help to assess the promise of the developed photocatalyst.
Round 2
Reviewer 2 Report
The authors have carefully revised the manuscript according to my recommendations. I would suggest the acceptance.